# Bioacaricides in Crop Protection—What Is the State of Play?

**DOI:** 10.3390/insects16010095

**Published:** 2025-01-17

**Authors:** Dejan Marčić, Ismail Döker, Haralabos Tsolakis

**Affiliations:** 1Laboratory of Applied Entomology, Institute of Pesticides and Environmental Protection, Banatska 31B, 11080 Belgrade, Serbia; 2Department of Plant Protection, Agricultural Faculty, Cukurova University, 01330 Sarıçam, Türkiye; idoker@cu.edu.tr; 3Department of Agricultural, Food and Forestry Sciences, University of Palermo, Viale delle Scienze 13, Ed. 5, 90128 Palermo, Italy; haralabos.tsolakis@unipa.it

**Keywords:** biopesticides, bioacaricides, plant-feeding mites, pest management

## Abstract

Bioacaricides (biological acaricides) are pesticides of biological origin used to protect crops from mite pests. There are three types of bioacaricides. Microbial acaricides are based on microorganisms (fungi and bacteria) that are pathogenic to mites. Other bioacaricides are manufactured from natural active substances of microbial, plant or animal origin and control mites by toxic action (biochemicals) and non-toxic action (semiochemicals). Among microbial acaricides, the largest group is mycopesticides, which are products based on living propagules of fungi. The most commercially successful biochemicals are products based on substances obtained from actinomycetes (a group of bacteria) and higher plants, mostly aromatic. Bioacaricides can be used in programs for the integrated management of mite pests. In order for bioacaricides to be included in these programs, it is necessary, among other requirements, to evaluate their compatibility with predators, which are the biological control agents of mite pests.

## 1. Introduction

Crops are threatened by a variety of pests—harmful organisms that reduce their productivity—such as animal pests (insects, mites, nematodes, molluscs), microbial pathogens (bacteria, fungi, viruses), and weeds [1]. Considering that crop losses due to pests can be substantial (higher than 80% in some crops), pest control measures are required. The use of synthetic chemical pesticides for this purpose has increased dramatically since the mid-20th century. The application of synthetic pesticides in many cases has improved crop yield and quality, boosted food security and increased farmers’ income [2,3]. On the other hand, the widespread adoption of synthetics has also led to negative outcomes, such as the development of pest resistance to pesticides, adverse effects on non-target organisms, environmental contamination and increased risks to human health [4,5].

Growing public demand for environmentally safe and sustainable pest management, along with increasingly stringent pesticide regulatory requirements, has boosted the interest in pesticides of biological origin—biopesticides—as alternatives to synthetic chemicals [6,7,8]. Biopesticides are not a novelty in pest control and crop protection. Several plant-based products (e.g., nicotine, pyrethrum, rotenone) were commercialized for use against animal pests in Western Europe and the United States during the 19th and early 20th centuries [9,10]. After World War II, development of biopesticides continued, albeit in the shadow of the large-scale production and use of synthetic chemical pesticides. The most important biopesticides included products based on the entomopathogenic bacterium *Bacillus thuringiensis* Berliner, fermentation products from soil actinomycetes, mycopesticides based on the entomopathogenic fungi *Beauveria bassiana* (Balsamo) Vuillemin and *Metarhizium anisopliae* (Metschnikoff), as well as azadirachtin and other plant-based products derived from the neem tree (*Azadirachta indica* A. Juss). Despite their ongoing development, biopesticides have played a minor role in crop protection. However, since the 21th century, the biopesticide sector has begun to experience faster growth in the global market compared to the sector of synthetic pesticides [6,7,11].

The family of spider mites (Tetranychidae) includes several major pests of crop plants, of which the most important are the two-spotted spider mite, *Tetranychus urticae* Koch, the citrus red mite, *Panonychus citri* (McGregor) and the European red mite, *Panonychus ulmi* (Koch). Economically important pests are also found in other mite families, such as Eriophyidae [the citrus rust mite, *Phyllocoptruta oleivora* (Ashmead)*,* and the coconut mite, *Aceria guerreronis* Kiefer], Tarsonemidae [the broad mite, *Polyphagotarsonemus latus* (Banks), and the cyclamen mite, *Phytonemus pallidus* (Banks)], and Tenuipalpidae (flat mites, *Brevipalpus* spp.).

Acaricides—pesticide products used against plant-feeding mites—remain a vital component of integrated pest management (IPM) programs in crop protection. It should be noted that a considerable number of active substances used against mites are actually insecto-acaricides, i.e., their spectra of toxic activity include both insects and mites. This overlap occurs because these substances typically act on molecular target sites common to both insects and mites [12,13,14,15]. In this review, we use the term acaricide to refer to all pesticides intended to control mites, regardless of whether they are also labeled for insect control. In the context of IPM, one of the crucial points is acaricide selectivity—i.e., their compatibility with predatory mites used as biological control agents (BCAs) of tetranychids and other plant-feeding mites. Among contemporary acaricides, there are not many products of biological origin. Considering recent trends in the global pesticide market, however, an increase in the number of acaricidal products of biological origin (bioacaricides) could be expected. This review will focus on the properties and effects of contemporary bioacaricides intended for use in crop protection. To carry out the review on bioacaricides, however, we first need to clarify what we actually refer to when we discuss biopesticides in the modern world.

## 2. Biopesticides in the Modern World

### 2.1. Definitions and Classifications

There is no global consensus on the definition of the term biopesticide (Figure 1). The Organisation for Economic Co-operation and Development (OECD) distinguished four types of biological pesticides as products used for pest control: microbial pesticides (microbials)—microorganisms (bacteria, fungi, viruses etc.) and any associated metabolites to which the effects of pest control are attributed; botanical pesticides (botanicals)—active substances obtained by processing plant material (plant extracts and essential oils, and/or their components); semiochemicals—chemical substances emitted by animals, plants, and other organisms that evoke a response of individuals of the same or other species (allelochemicals, pheromones); and invertebrates—insects, mites and other animals used as the BCAs used in pest control [16,17,18,19]. The Food and Agriculture Organization of the United Nations (FAO) and the World Health Organization (WHO) define biopesticides as substances derived from nature that may be formulated and applied in a manner similar to chemical pesticides. This definition includes microbials, botanicals and semiochemicals, but not invertebrates [20].

On the other hand, the Environmental Protection Agency of the United States (US EPA) distinguishes three classes of biopesticides (defined as pesticides derived from natural materials): microbial pesticides (microorganisms as active ingredients); biochemical pesticides (natural substances that control pests by non-toxic action) and plant incorporated protectants (pesticidal substances that plants produce from genetic material that has been added to the plant). Biochemical pesticides include insect pheromones, growth regulators, repellents, attractants, induced resistance promoters, suffocating agents and desiccants [21,22]. The US EPA has established a special committee to determine which substances meet the criteria to be classified as biochemical pesticides. Considering their toxic action, microbial metabolites produced at the industrial scale (e.g., fermentation products of soil actinomycetes) and some botanicals (e.g., pyrethrum) are not included in biochemical pesticides. This class corresponds to semiochemicals and a part of botanicals from the OECD and FAO-WHO definitions (Figure 1), with the exception of substances of animal origin and some other natural substances.

In China, biopesticides are divided into five types: microbial pesticides, plant-derived pesticides, biochemical pesticides, agricultural antibiotics and natural enemies [23]. The first, second and fifth types match the OECD’s first, second and fourth types, while agricultural antibiotics match microbial metabolites produced at the industrial scale (Figure 1). Biochemical pesticides are defined similarly to the US EPA’s classification, with the difference that plant-based growth regulators (e.g., azadirachtin) are included in the second type.

In the European Union (EU), biopesticides are not formally recognized as a distinct category. EU legislation recognizes microorganisms as one type of active substance used in plant protection products [24]. In addition to microorganisms, the EU Pesticide Database [25] includes active substances that can be classified as botanicals, semiochemicals, or microbial metabolites produced at the industrial scale (Figure 1). The types of low-risk substances and basic substances are also recognized in the EU legislation on plant protection products [24], and many of these substances belong to biopesticides [26].

According to a widely accepted definition proposed by Bailey et al. [27] and Chandler et al. [28], biopesticides are mass-produced and biologically based pest control products that fall into three main categories: microorganisms (microbial biopesticides), semiochemicals and biochemicals (natural substances of microbial, plant and animal origin with toxic action). The last category, with the exception of few animal-derived biochemicals, matches botanicals and microbial metabolites produced at the industrial scale (Figure 1). Taking into account what is common to all the above definitions, biopesticides can be broadly defined as pesticides manufactured from living organisms and their biologically active products.

### 2.2. Global Market, Advantages and Constraints

Due to the lack of a globally agreed-upon definition of biopesticides, as well as various methods of data collection and processing, estimates of the global biopesticide market can vary widely. An additional challenge arises from the difficulty in separating data on biopesticide use in public health programs from its use in crop protection. Nevertheless, it is clear that the biopesticide sector represents a small portion of the global pesticide market, as well as the fact that this sector has been growing rapidly: from a mere 1% of the global market in 1998 to approximately 10% in 2022 [6,8,29]. This increased adoption of biopesticides is attributed to technological advancements in this sector, as well as to restrictions on the use of synthetic pesticides, the broader implementation of IPM programs and rising demands for organic food in developed countries [8,30].

Low mammalian toxicity, zero or low harvest and re-entry restrictions, higher safety for human health, non-target organisms and the environment, as well as lower risk of target pests developing resistance, have been the most frequently highlighted advantages of biological pesticides over the synthetic chemical ones [6,28]. Although there are good grounds for such a general perception, there are also data indicating that a comparison of biopesticides and synthetic pesticides is not simply a black-and-white issue. Notable examples include the toxic and adverse sublethal effects of the fermentation products of soil actinomycetes (abamectin, spinosad), botanicals (essential oils, neem-based products) and mycopesticides on pollinators and predators and parasitoids of insect and mite pests [31,32,33,34], as well as the harmful effects of fungal biopesticides on mycorrhizal and saprophytic fungi, soil bacteria and plants [35]. Another often neglected issue is that biopesticides are also vulnerable to resistance evolution. According to the Arthropod Pesticide Resistance Database (APRD), 28 insect pest species and 19 insect and mite pest species have developed resistance to spinosad and abamectin, respectively [36]. Resistance to antibiotics (another group of fermentation products of actinomycetes) has been recorded in some plant pathogenic bacteria [37], while several lepidopteran pest species have developed resistance to *Bt*-toxins [38,39].

The further expansion of the global biopesticide market depends on the successful overcoming of various constraints. The most important technical constraints include lower and variable efficacy of biopesticide products and their greater susceptibility to unfavorable environmental conditions compared to the synthetic pesticides, as well as the issue of quality and availability of resources [28,30]. From a socio-political point of view, regulatory barriers and inconsistent policies and regulatory procedures across regions, along with limited awareness of biopesticides among the end-users [8,30,40], are the major constraints to the wider use of biopesticides.

## 3. Bioacaricides in Contemporary Crop Protection

Within the extensive recent scientific literature on biopesticides, there have been few reviews specifically dealing with bioacaricides in crop protection. Flamini [41] published a comprehensive review of plant-derived compounds used against harmful species from the subclass Acari in agriculture, veterinary and human medicine, covering the last decade of the 20th century. Other reviews have mainly focused on the use of bioacaricides against *T. urticae*. Maniania et al. [42] reviewed the use of fungal pathogens in the inundative, conservational and classical biological control of *T. urticae* and *Tetranychus evansi* Baker and Pritchard, the red tomato spider mite. Attia et al. [43] and Rincón et al. [44] reviewed the research on botanicals, while Jakubowska et al. [45] presented examples of plant extracts, microorganisms and predatory mites used in the control of *T. urticae*.

Below we provide an overview of the properties, effects and uses of contemporary bioacaricides. They are defined as commercialized crop protection products used in the control of plant-feeding mites and can be categorized as follows: based on microorganisms (***microbial acaricides***); biochemical active substances with toxic action (***biochemical acaricides***) and with non-toxic action (***semiochemicals***). Furthermore, biochemical acaricides are further divided into *microbial biochemical acaricides* and *botanical biochemical acaricides* based on their origin. Special emphasis will be given to their compatibility with predatory mites from the family Phytoseiidae, which serve as biological control agents in the integrated management of plant-feeding mites and insects.

### 3.1. Microbial Acaricides

Microbial biopesticides account for around 60% of the global biopesticide market, and they are dominated by products based on *δ*-endotoxins (crystal proteins) from *B. thuringiensis* (*Bt*) [8,46]. The spectrum of activity of these products includes lepidopteran, coleopteran and dipteran insect pests, but not mites. Some *Bt* strains are capable of producing non-proteinaceous *β*-exotoxins, such as thuringiensin. Besides insecticidal activity, thuringiensin also shows considerable acaricidal activity against spider mites [47,48,49]. Due to the mode of the toxic action of *β*-exotoxins (the inhibition of DNA-dependent RNA polymerase) which may affect mammals, their absence is a requirement for *Bt* products in Europe, the U.S. and Canada. In some countries, such as Russia, *β*-exotoxin-containing products are still used [50,51,52].

In the shadow of *Bt*, two betaproteobacterial species have recently been used by Marrone Bio Innovation (U.S.A.) as the basis for the development of crop protection products with insecticidal and acaricidal activity. The betaproteobacterium *Chromobacterium subtsugae* Martin et al., isolated from forest soil in the USA, is the source of the commercial product “Grandevo”, which contains the strain PRAA4-1T. This product is labeled for controlling insect pests from the orders Lepidoptera, Hemiptera and Thysanoptera, as well as plant-feeding mites. During fermentation, the bacterium produces violacein and other secondary metabolites that contribute to its toxicity [8,53]. Another commercially successful betaproteobacterial species is *Burkholderia rinojensis* Cordova-Kreylos et al., isolated from a soil sample in Japan (strain A396). Its commercial product “Venerate” is labeled for use against several hemipteran species and plant-feeding mites. Various bacterial metabolites formed during fermentation are involved in its toxic effect [8,54]. In addition to the U.S., where these products were developed, they have also been registered in Canada, Mexico, Chile, New Zealand and some other countries. Both products contain inactivated bacterial cells and spent fermentation media. These are slow acting products that take several days to achieve their full toxic effect. Therefore, they should be used early, when mite populations are still at a low level, with consecutive applications recommended on a 7-day schedule, preferably rotated with other products. Laboratory bioassays have shown that “Grandevo” exhibits low to moderate toxicity to *T. urticae* female adults [55,56,57], as well as a significant reduction in fecundity [57]. The product “Venerate” was also moderately toxic to females [56]. Field evaluations showed moderate effectiveness of “Grandevo” in controlling *P. latus* on pepper and *P. oleivora* on orange trees [58,59], as well as *T. urticae* on strawberry [60].

Among microbial acaricides, mycopesticides (insecticides and acaricides) manufactured from living propagules of entomo- and acaropathogenic fungi [61,62] form a large group of commercial products. However, it should be mentioned that the majority of these are based on two fungal species, *B. bassiana* s.l. and *M. anisopliae* s.l. Other commercially important species include *Isaria fumosorosea* Wize, *Lecanicillium muscarium* (Petch) Zare & Gams, *Lecanicillium lecanii* (Zimmermann) Gams and Zare and *Hirsutella thompsonii* Fisher [46,63,64]. Their pathogenic mode of action against insects and mites is based on the penetration of the host integument (by mechanical force and cuticle degrading via hydrolytic enzymes), as well as colonization and the production of toxic metabolites, leading to mycosis and death. Successful application of mycopesticides relies particularly on relative humidity (RH): their efficacy increases with raising RH [65,66,67].

The majority of mycopesticides are intended for use in horticultural crops, including greenhouse and field vegetables, ornamentals and fruit crops. Only a smaller portion of these products is labeled for controlling both insects and mites, while the vast majority are registered solely as mycoinsecticides. For example, more than 20 strains of *B. bassiana* have been used for manufacturing mycoinsecticides [63,64], yet less than one-third of them are also registered as mycoacaricides (Table 1). The target pest range of these mycoacaricides varies. Some products are labeled only for the control of *T. urticae* and other tetranychid mites, while others are also used against Tarsonemidae (*P. latus*, *P. pallidus*), Eriophyidae (*P. oleivora*) and Tenuipalpidae. Certain products are labeled for use against all plant-feeding mites. Unlike most mycopesticides, the products based on *H. thompsonii*, a mite-specific fungus, are registered exclusively as mycoacaricides in India, targeting *A. guerreronis* and various spider mites [68]. In addition, there are also some differences between countries in the intended use of products derived from the same fungal strains. For example, products based on the strains ATCC 74040, GHA and PPRI 5339 of *B. bassiana* and the strains Apopka 97 and FE9901 of *I. fumosorosea* are registered for controlling sucking insect pests and plant-feeding mites in the U.S.A., while in the EU they are not intended to use as mycoacaricides. The only mycoinsecticide and mycoacaricide registered both in the EU and in the U.S.A. is a product based on *Metarhizium brunneum* (Petch) strain F52 [25,69].

Field and greenhouse trials have shown moderate to high effectiveness of mycopesticide products in controlling tetranychid and other plant-feeding mites in various countries and regions. Typically, at least two treatments at 7–14 day intervals are required for effective control, which may prove uneconomical unless used in higher value crops. The variable effectiveness of the mycopesticides is largely attributed to their high sensitivity to ultraviolet radiation and other environmental conditions, which is less pronounced in greenhouses [75,76,77,78,79,80,81]. In addition to appropriate formulations, the sustainability of mycopesticide use in the control of plant-feeding mites can be improved by including them in IPM programs. Within these programs, mycopesticides can be combined with other bioacaricides or synthetic acaricides, as well as predatory mites and insects [42,46,63,82].

### 3.2. Biochemical Acaricides

#### 3.2.1. Microbial Biochemical Acaricides

Secondary metabolites produced by fermentation from soil actinomycetes have been one of the most important sources for the mass production of biopesticides. Products based on avermectins and milbemycins, macrocyclic lactones obtained from *Streptomyces avermitilis* (Burg et al.) Kim and Goodfellow and *Streptomyces hygroscopicus* subsp. *aureolacrimosus* Takiguchi et al., respectively, are very well-known examples of biopesticides that are commercially successful at the global level [6,12]. Abamectin (a mixture of avermectins B_1a_ and B_1b_) and milbemectin (a mixture of milbemicins A_3_ and A_4_) are used against tetranychid, eriophyid and tarsonemid mite pests. Abamectin is also labeled for use against a broad range of insect pests, while milbemectin is primarily used against mites. They have a neurotoxic mode of action—specifically, allosteric modulation of glutamate-gated chloride channels, causing paralysis of mites and insects [15]. Abamectin is marketed under a wide variety of trade names, such as “Agri-Mek”, “Dynamec” and “Vertimec” from Syngenta, the largest manufacturer. Mitsui Chemicals, the company that developed milbemectin, sells it under the trade names “Milbeknock” and “Koromite”. Together, abamectin and milbemectin occupy third place in the global market for products targeting insects and mites [83]. Despite being sensitive to photolytic degradation, these biochemical acaricides provide residual activity, due to rapid penetration into the leaves, and translaminar movement. Their residual action against mites can be further improved by mixing with various spray adjuvants [59,84,85].

Another globally successful biopesticide is spinosad, a mixture of spinosyns A and D, derived from the fermentation of actinomycete *Saccharopolyspora spinosa* Mertz and Yao. It is also a neurotoxic compound, acting as an allosteric modulator of nicotinic acetylcholine receptors in the insect central nervous system [15]. Spinosad is highly effective against a wide range of insect pests, such as thrips, caterpillars and leaf miners [7]. Some of its products are also labeled for spider mite control, such as “Conserve SC” (Corteva Agriscience). The recommended application rate for controlling spider mites with this product (180 g a.i/l) is 2–3 times higher than those for insect control. The acaricidal properties of spinosad have been supported by several bioassays. In acute toxicity bioassays with *T. urticae* females [66,86,87,88], the LC_50_ estimations were found to be far below the recommended rate. Van Leeuwen et al. [86] demonstrated the possibility for the systemic use of spinosad to control spider mites, applied to the roots of tomato plants grown in rockwool. The instructions for using the product labeled for spider mites recommend an early application (before mite populations have become abundant), reapplication after several days and the addition of a nonionic adjuvant to improve spray coverage. Poor spray coverage and late application may, at least partially, explain the variable results of spider mite controls in field trials with spinosad [89,90].

Due to their toxic action, the US EPA does not consider abamectin, milbemycin and spinosad as biopesticides. Also, they are not addressed in the OECD and FAO-WHO guidelines on biopesticides [17,20,22,91]. On the other hand, Bailey et al. [27] and Chandler et al. [28] considered them biochemical pesticides, and they are classified as biopesticides in India and China [23,30].

#### 3.2.2. Botanical Biochemical Acaricides

The history of biopesticides of plant origin (botanicals) intended to control insects and mites has been largely a tale of two plants: the Dalmatian daisy, *Tanacetum cinerariifolium* (Trevir.) Sch. Bip. (Asteraceae), and the Indian neem tree, *Azadirachta indica* A. Juss. (Meliaceae). The most widely used botanicals are commercial products that have come from these two plant species. Since the early 2000s, essential oil-based products have also gained increasing importance in the market of botanicals [10,40,62].

Dried flowers of *T. cinerariifolium* are a source of oleoresin pyrethrum, a crude flower dust containing a mixture of six related esters as active ingredients. The most important ones are pyrethrin I (an ester of pyrethrolone and chrysanthemic acid) and pyrethrin II (an ester of pyrethrolone and pyrethric acid). Technical grade pyrethrum used in formulating commercial products contains 20–25% pyrethrins [10]. Pyrethrins are neurotoxic: they act as sodium channel modulators that keep the channels open and cause hyperexcitation [15].

Although it has been on the market for more than a century, pyrethrum is still one of the leading botanicals, partly due to its non-agricultural use (structural pest control, public health), and because of its relatively low toxicity to humans and mammals [92,93]. Eastern African countries had long been world leaders in the cultivation of *T. cinerariifolium* and pyrethrum production before Australia (Tasmania) took the lead in the past two decades [40,94]. Pyrethrum-based products are recommended for use against a range of insect and mite pests in home gardens, as well as in commercial crop and ornamental production (Table 2).

With regard to *A. indica,* its seeds are a rich source of neem oil, which contains numerous active compounds, including the four most important tetranortriterpenoids (limonoids): azadirachtin, salannin, meliantriol and nimbin. Azadirachtin is a slow acting insect and mite growth regulator and a potent feeding and oviposition deterrent. Its exact mode of action on the growth and developmental processes at the molecular level is still unclear [97,98,99,100,101]. Various types of products based on azadirachtin and neem oil are currently available on the market. In addition to insect pests, most related products are also labeled for the control of mites, primarily tetranychids (Table 2). A plethora of these products have been registered in India, where the neem tree is native. The current global status of azadirachtin-based biopesticides is primarily due to the registration of a great number of such products in many developing countries of Africa and Latin America (as a consequence of the introduction of neem trees), as well as in China [23,40,97]. Azadirachtin-based products have shown moderate to high efficacy in controlling spider mites and other plant-feeding mites [102,103,104,105]. The formulation type and production technology have a great influence on their effectiveness [10].

A great increase in research interest in botanicals over the past two decades has resulted in moderate success in the commercialization of new products intended to control insect and mite crop pests [40,94,96]. Far behind pyrethrum and azadirachtin/neem, the third place in the global market is held by crop protection products based on plant essential oils and/or their biologically active constituents. Essential oils, complex mixtures of volatile secondary metabolites, are obtained via distillation from aromatic plant species belonging to the families Lamiaceae, Myrtaceae, Rutaceae and several others. Terpenoids (monoterpenes and sesquiterpenes) and, to a lesser extent, phenylpropanoids are the main constituents of essential oils. Some terpenoids can also be obtained via economically viable industrial synthesis. Essential oil-based commercial products contain either a single oil/constituent or a mixture of oils/constituents. The products act rapidly by direct contact and the vapor phase. Owing to their volatile nature, they are effective fumigants in closed spaces. Their lethal effects on insects and mites are most likely a consequence of neurotoxic action of the constituents on one or more molecular targets, such as acetylcholinesterase, octopamine receptors and GABA-gated chloride channels. Given the complex mixture of terpenoids in most essencial oils, their toxic action appears to be a result of a synergy among their constituents. In addition to the lethal effect, essential oil-based products also cause repellent and deterrent effects [10,40,94,96,106].

The most common essential oils that have been used for manufacturing commercial products intended to control mite pests are rosemary oil, peppermint oil, orange oil, tea tree oil and cinnamon oil (Table 2). The acaricidal activity of the essential oil-based products is based on their constituents, such as monoterpenes carvacrol, 1.8-cineole, citronellol, *d*-limonene, eugenol, geraniol, menthol and phenylpropanoid cynnamaldehyde. One terpenoid can have several botanical sources [40,96]. It is interesting to note, as Isman [94] pointed out, that the US EPA includes within the category of biochemical biopesticides, defined as non-toxic substances, several monoterpenes (citronellol, eugenol, geraniol, methol) and essential oils (eucalyptus oil, orange oil, tea tree oil) even though these are clearly neurotoxic to insects and mites.

Essential oil-based products are usually recommended for use against a range of insect and mites, primarily tetranychids. In some products, essential oil is combined with various non-volatile plant (vegetable) oils, such as castor oil and cottonseed oil. Some of these products (“Akabrown”, “Mitexstream”, “Biomite”) are only labeled for the control of mite pests. Greenhouse and field evaluations have shown that these products can control mites as effectively as other botanical biochemical acaricides, and even as well as synthetic chemical acaricides [107,108,109,110]. Considering their low persistence, repeated treatments are usually recommended. On the other hand, the extended efficacy of essential oil-based products observed in some field trials suggests that their behavioral effects may contribute considerably to the overall impact [96].

One of the most important recently introduced biopesticide products is “Requiem”, which is based on monoterpenes *α*-terpinene, *d*-limonene and *p*-cymene as active ingredients (Table 2). It is a contact insecticide and acaricide with strong repellent and deterrent activity [111,112,113]. The product was initially manufactured from the extract of American wormseed, *Chenopodium ambrosioides* L. near *ambrosioides* (Chenopodiaceae), containing these three monoterpenes as the major constituents. Since Bayer Crop Science took over the product, it has been made from a blend of synthesized monoterpenes, intended to mimic the naturally-occurring extract [40,96]. The product is considered apt for use against a broad range of plant-feeding mites and insects, mostly hemipteran.

Some newer botanical biochemical acaricides have alkaloids as active ingredients (Table 2). Products based on the extracts from *Sophora flavescens* Aiton (Fabaceae), a Chinese medicinal herb, contain matrine, oxymatrine and other quinolizidine alkaloids. Besides insect pests, these products have proven effective against spider mites and flat mites [78,101,114,115,116]. Veratrine and related cevadine-type alkaloids are active ingredients in the products extracted from *Veratrum nigrum* L. (Melanthiaceae), an herb also used in Chinese traditional medicine. Another recent product is “Captiva”, which is based on a combination of canola oil and garlic oil with capsaicin, an alkaloid found in the oleoresin of red pepper, *Capsicum annuum* L. (Solanaceae). The product is labeled for use against spider mites, *P. oleivora*, *P. latus*, *Brevipalpus phoenicis* (Geijskes) and a broad range of insect pests [40,94,96].

In addition to their combining with essential oils, vegetable oils are also used independently to formulate commercial biopesticide products. Canola oil and rapeseed oil are two vegetable oils widely used as biopesticides (Table 2). They come from the same botanical source (*Brassica napus* L., *Brassica rapa* L.), but canola oil is produced from cultivars with a low content of erucic acid and glucosinolates. The main active constituents of these oils are fatty acids, such as oleic acid and linoleic acid [7,117]. Oleic acid is also the main constituent of both cottonseed oil and safflower oil, while ricinoleic acid is the main constituent of castor oil (Table 2). Prevention of gas exchange (suffocation) is the widely accepted theory for the major mode of action of plant oils against soft-bodied insects and mites, while the interference of fatty acids with the cell membrane constituents, leading to its disruption, is of lesser importance [7]. However, Takeda et al. [118] indicated the inhibition of larval hatching instead of suffocation as the major mode of action of plant oils against *T. urticae* eggs. Canola oil and rapeseed oil are also used to formulate commercial biopesticide products in combination with other botanicals, such as pyrethrum and azadirachtin/neem [23,25,69].

Botanical biochemical acaricides can reduce mite populations to levels comparable to those achieved by synthetic chemical acaricides. However, due to the low persistence of botanicals, achieving such results requires the application of relatively high concentrations and repeated treatments. A tank mix application of botanical biochemical acaricides with synthetic chemical products is one of the ways to increase their effectiveness. Another way is to combine two botanicals, or botanicals and microbials. In order to achieve the long-term and sustainable use of biopesticides, it is necessary to include them into IPM programs [40,95,96,100].

### 3.3. Semiochemicals as Bioacaricides

Semiochemical biopesticides include commercial crop protection products based on behavior-modifying signaling chemicals of plant or animal origin that carry information between individuals and are intended for the control of insect and mite pests. They are divided into pheromones, inducers of behavioral response in the receiving individuals of the same species, and allelochemicals, which are involved in interspecific interactions [28]. The most widely used semiochemical biopesticides are insect sex pheromones: a lot of pheromone-based products are applied in mating disruption, mass trapping, attract-and-kill, and other control strategies [119].

Unlike the situation with insects, sex pheromones of plant-feeding mites have negligible importance as biopesticides. Nerolidol and farnesol, two structurally related sesquiterpenes found in immature *T. urticae* females, act as sex pheromones to attract males for mating [6,120]. They are also constituents of some plant essential oils. As active ingredients of the product “Biomite” (Table 2), nerolidol and farnesol increase the activity of mites and consequently their exposure to the co-formulated toxic monoterpenes geraniol and citronellol.

Allelochemicals are divided into allomones, kairomones and synomones, depending on whether the response is favorable to the emitter, the receiver or both. Plants produce various secondary metabolites (allomones) to repel or deter insects and mites. The repellent effects of essential oils and the feeding deterrent effects of azadirachtin/neem are well known, but these effects are of secondary importance to the control of crop pests [96,100]. There have been no commercially successful products based unequivocally on repellency or feeding deterrence as the major activity. Kairomones are volatile plant secondary metabolites, induced by plant-feeding insects and mites, that attract their predators and parasitoids as receivers. Methyl salicylate, one of the best-known attractants, is synthesized and available as a slow-release dispenser (commercial product “PredaLure”, manufactured by AgBio, Westminster, CO, USA). The product demonstrated the effective attraction of phytoseiid predators in several field trials [120].

## 4. Side Effects of Bioacaricides on Predatory Mites

The IPM paradigm in crop protection defines pesticide selectivity as the compatibility of pesticides with the natural enemies of plant-feeding insect and mite pests, and it is based on the evaluation of the detrimental effects of pesticides on parasitoid and predatory insects and mites as the BCAs of the pests [121]. The most important BCAs of spider mites and other plant-feeding mites, as well as some insect pests (whiteflies, thrips), are predatory mites of the family Phytoseiidae. Among the more than 30 commercialized phytoseiids, the major species are *Amblyseius swirskii* Athias-Henriot, *Neoseiulus californicus* (Garman), *Neoseiulus cucumeris* (Oudemans) and *Phytoseiulus persimilis* Athias-Henriot, which together account for 60% of the global market [122]. The selectivity of pesticides with the BCA has two aspects. Physiological selectivity is based on toxicokinetic and toxicodynamic mechanisms that ensure the lower sensitivity of the BCAs compared to harmful species. Ecological selectivity is the result of limiting the exposure of the BCA to pesticides in time and space [121,123].

Various methods have been used for the evaluation of pesticide selectivity. Since the 1970s, the Working Group (WG) “Pesticides and Beneficial Organisms” of the International Organization for Biological Control—Western Palearctic Regional Section (IOBC-WPRS) has developed standardized methods for over 30 beneficial insect and mite species, parasitoids and predators of insect and mite pests, and has tested nearly 400 pesticides [124,125]. The IOBC’s methods constitute a program for the sequential testing of pesticide effects on beneficial organisms, in which the decision to perform field trials depends on the outcome of laboratory bioassays. A harmless classification implies that further testing is not necessary—i.e., the pesticide is considered compatible [126,127]. An initial toxicity bioassay is carried out on glass plates, exposing the most susceptible life stage to fresh deposits of pesticides applied at the highest recommended rates (the “worst-case” scenario). The other laboratory bioassays include modifications such as exposure of less susceptible stages on leaves or leaf discs (extended laboratory bioassay) and exposure to field aged pesticide residues (persistence bioassay) [124,128].

In laboratory bioassays, the IOBC ranking for classifying harmfulness is based on the reduction in beneficial capacity as a consequence of mortality and reduced fecundity. It is expressed by the coefficient of toxicity or the total effect (*E*, %), calculated using the formula *E* = 100% − (100% − M) × *R*, where *M* is the percentage of mortality corrected for mortality in the control and *R* is the ratio between the number of eggs produced by treated females and the number of eggs produced by females in the control [129]. Instead of the number of eggs laid (fecundity), in some studies [130,131,132] the number of hatched eggs (fertility) was recorded.

The IOBC ranking includes four categories, based on the values of *E* (%): 1 = harmless (*E* < 30%), 2 = slightly harmful (*E* = 30–80%), 3 = moderately harmful (*E* = 80–99%) and 4 = harmful (*E* > 99%). If a pesticide is classified in categories 2–4, further testing in semi-field and field trials is necessary. To calculate *E* in these trials, the effects of pesticides on population size and dynamics are recorded, and a pesticide is considered harmless (*E* < 25%) slightly harmful (*E* = 25–50%), moderately harmful (*E* = 50–75%) or harmful (*E* > 75%) [128].

The IOBC-WPRS database on the effects of plant protection products on beneficial arthropods (set up in the early 2000s, currently under revision) compiled data concerning the compatibility of pesticides and beneficials that were published in scientific journals and proceedings of the WG conferences, as well as in the reports on regulatory testing in the European Union. The database includes 1768 test results on the selectivity of 379 pesticides to 16 phytoseiid species [125,133]. Acaricides are represented, with 567 results for 98 compounds. Besides acaricides, other pesticides have also been evaluated, including 143 herbicides and 111 fungicides, as well as 27 insecticides. With 346 tested pesticides (76% of the total results and 60% of the acaricide results), the species *Typhlodromus pyri* Scheuten stands out among the phytoseiids, followed by *P. persimilis*, *Amblyseius andersoni* (Chant) and *Euseius finlandicus* (Oudemans). The predominance of *T. pyri* is mainly a consequence of its status as an indicator species, since over 86% of the results originate from reports on the regulatory testing of pesticides in the European Union [133]. The database contains test results of 11 bioacaricides and 10 phytoseiid mites, with a total of 65 results (55 from laboratory bioassays and 10 from field trials), 69% of which are for *T. pyri*. Among bioacaricides, more than a half of the results refer to abamectin, spinosad and *B. bassiana* (Table 3a).

In laboratory bioassays, abamectin was shown mostly to be moderately harmful to harmful, with the expected variation in results depending on predator species and/or abamectin concentrations. The persistence bioassay showed that compatibility could be achieved by exposure of *N. californicus* to residues aged at least 15 days. In field trials, abamectin was slightly harmful to *A. swirskii* and harmless to *T. pyri*. Spinosad results were similar to abamectin results, while *B. bassiana* was slightly harmful or harmless to *T. pyri* and other phytoseiids (Table 3b).

Most of the data on the compatibility of acaricides with phytoseiid mites were not derived from the IOBC-WPRS database. A number of papers were not included in the database because they used methods that deviated more or less from the standard characteristics [124,125]. The experimental designs of many other studies, carried out almost exclusively on leaves or leaf discs, have included new species of phytoseiid mites, different developmental stages, a greater number of acaricide concentrations, new ways of exposure and new toxicity parameters and endpoints [123,134]. Concentrations lower than the recommended ones have also been tested, and some of the studies have been carried out to estimate LC_50_ and other LCs. In addition to mortality for eggs, juveniles and/or adults, many studies include an assessment of the sublethal effects of pesticides on developmental time, fecundity, fertility and other life history traits of phytoseiid mites. Different methods of exposure to pesticides have been applied in the bioassays individually or in combinations, such as triple exposure: a combination of direct treatment, residual exposure and feeding of predators with treated prey. Some studies also included an evaluation of the comparative toxicity between predator and prey [130,135,136,137,138]. In addition to the total effect and the IOBC classification of toxicity, new parameters and criteria for compatibility evaluations were introduced. Beers and Schmidt [139] and Schmidt-Jeffris and Beers [140] proposed that the effect of acaricides is expressed by calculating the cumulative effect on the survival of treated females, their fecundity and fertility (egg hatching), as well as on the survival of hatched larvae. For predator-prey comparisons, the selectivity index is calculated as the difference between the cumulative effects of acaricides on predator and prey.

Recently, the evaluation of the compatibility of pesticides with phytoseiid mites has increasingly been based on laboratory bioassays that estimate their effect at the population level. In the demographic bioassay, the effect is expressed by a change in the value of the intrinsic rate of population increase (*r_m_*) as a consequence of the pesticide impact on developmental time, fecundity, longevity and other life history traits of the mites. The demographic bioassay is based on construction of the life table from data on survival and reproduction. Two types of the life table have been used: the female fertility life table, constructed from data on female survival and fertility (production of female offspring) [141,142], and the age-stage two-sex life table, which includes juvenile development and survival data for both sexes, as well as female fecundity [143].

Examples of the aforementioned studies are presented in Table 4, Table 5, Table 6 and Table 7. Most of the results of these studies refer to abamectin, spinosad and azadirachtin. In addition to the four major commercialized species of global importance (*A. swirskii*, *N. californicus*, *N. cucumeris* and *P. persimilis*), these studies have included other important phytoseiid species, such as *Galendromus occidentalis* (Nesbitt), *Neoseiulus womersleyi* (Schicha), *Phytoseiulus macropilis* (Banks) and *Phytoseiulus longipes* Evans. Both commercialized and native strains have been evaluated. Mycopesticides based on *B. bassiana* and *H. thompsonii* were mostly compatible with phytoseiids (Table 4).

Abamectin and milbemectin proved to be not compatible after direct treatment, followed by residual exposure and consumption of treated prey. Abamectin was considered compatible mostly when predatory mites were exposed to its aged residues. This is an example of ecological selectivity that can be achieved by temporal separation between acaricides and phytoseiids [123]. Spinosad results were variable; similar to abamectin, its low persistence allowed ecological compatibility (Table 5).

**Table 5 insects-16-00095-t005:** Examples of evaluations of the compatibility of microbial biochemical acaricides and phytoseiid mites in various laboratory bioassays; Comp = compatibility conclusion: + positive (compatible); − negative (not compatible); → further research needed.

Bioacaricides	Phytoseiid Mites		Methodology		Comp	References *
			Exposure	Endpoints		
abamectin	*Amblyseius swirskii*	n	rc, rc+/−; Td-re	Se, Sl, Sf, Fec, LC_50_	→	[146]
	*Euseius scutalis*	n	rc, rc+/−; Td-re	Se, Sl, Sf, Fec, LC_50_	→	[147]
	*Galendromus occidentalis*	c	rc; Tre (3–37)	Sf, Fec, Fer, *E*, *IOBC*	+ Tre (6)	[148]
	*Neoseiulus barkeri*	c	rc, rc+/−; Tre (0)	Sf, Fec, Fer, LC_50_, *E*, *IOBC* ●	+	[149]
	*Neoseiulus californicus*	c	rc; Tre (0–21)	Sf	+ Tre (7)	[150]
		n	rc; Tre (0–21)	Sf ●	+ Tre (7)	[151]
		c	rc; Td-re + Tp	Sf, Fec, Fer, *SI* ●	−	[138]
	*Neoseiulus cucumeris*	n	rc, rc+/−; Tre (0)	Sf, LC_50_ ●	+	[152]
		c	rc; Tre (0)	Sf	−	[153]
	*Neoseiulus fallacis*	c	rc; Td-re + Tp	Sf, Fec, Fer, *SI* ●	−	[138]
	*Phytoseiulus longipes*	n	rc; Td-re	Sf, Fec, Fer, *E*, *IOBC*	−	[145]
			rc; Tre (4–31)		+ Tre (10)	
	*Phytoseiulus persimilis*	c	rc; Tre (3–37)	Sf, Fec, Fer, *E, IOBC*	+ Tre (14)	[148]
		n	rc; Td-re	Sf, Fec, Fer, *E*, *IOBC*	−	[131]
		c	rc; Tre (0–21)	Sf	+ Tre (14)	[150]
		c	rc; Td-re + Tp	Sf, Fec, Fer, *SI* ●	−	[138]
milbemectin	*Neoseiulus womersleyi*	n	rc; Td-re	Sel-a, Sf, Fec ●	−	[154]
	*Phytoseiulus persimilis*	n	rc; Td-re	Sel-a, Sf, Fec ●	−	[155]
spinosad	*Amblyseius swirskii*	n	rc, rc+/−; Td-re	Se, Sl, Sf, Fec, LC_50_	−	[156]
	*Galendromus occidentalis*	n	rc; Td-re + Tp	Se, Sf, Fec	+	[136]
	*Kampimodromus aberrans*	n	rc; Tre (0)	Sf, Fec, Fer, *E*	−	[157]
	*Neoseiulus cucumeris*	c	rc; Tre (0–6) + Tp	Sf, *IOBC*	+ Tre (4)	[158]
	*Neoseiulus fallacis*	n	rc, rc+/−; Td-re + Tp	Se, Sf, Fec, LC_50_	→	[137]
		n	rc; Tre (0), Tp	Sel-a, Sf, Fec	−	[159]
	*Phytoseiulus persimilis*	n	rc; Td-re	Sf, Fec, Fer, *E*, *IOBC*	+	[131]
	*Transeius montdorensis*	c	rc; Tre (0–6) + Tp	Sf, *IOBC*	+ Tre (5)	[158]

c = commercial strains, n = native populations; rc = recommended concentrations; rc+/− concentrations higher/lower than rc; Td-re = direct (topical) treatment and residual exposure; Tre (0) = exposure to fresh residues; Tre (x–y) = exposure to residues aged from x to y days; Td = direct (topical) treatment and transfer to untreated surface; Tp = consumption of treated prey; Se = egg hatching; Sel-a = egg hatching and larval survival to the adulthood; Sl = larval survival; Sl−a = larval survival to the adulthood; Sf = adult female survival; Fec = fecundity (eggs produced by treated females); Fer = fertility (hatched eggs produced by treated females); *E* = the coefficient of toxicity [129]; *IOBC* = classification of toxicity according the IOBC-WPRS guidelines [128]; *SI* = selectivity index; ● comparative toxicity predator—prey; * Search of the literature (article written in English, published after the year 2000) was carried out by using combinations of several keywords in GoogleScholar, such as biopesticide, actinomycetes, Tetranychidae, Phytoseiidae, compatibility, effects.

Azadirachtin was shown to be compatible with phytoseiids, regardless of the route of exposure, while pyrethrum was not compatible after direct treatment and residual exposure. Similar to abamectin and spinosad, the low persistence of oxymatrine allowed for its compatibility (Table 6). In research that includes LC_50_ as one of the bioassay endpoints [137,146,147], conclusions were made based on the comparison of the LC_50_ values with the recommended concentrations.

**Table 6 insects-16-00095-t006:** Examples of evaluation of the compatibility of botanical biochemical acaricides and phytoseiid mites in various laboratory bioassays; Comp = compatibility conclusion: + positive (compatible); − negative (not compatible).

Bioacaricides	Phytoseiid Mites		Methodology		Comp	References *
			Exposure	Endpoints		
azadirachtin	*Amblyseius andersoni*	n	rc, Td-re	Sf, Fec, *E*	+	[160]
	*Neoseiulus barkeri*	c	rc, Tre (0)	Sf, Fec, Fer, *E*, *IOBC*	+	[149]
	*Neoseiulus californicus*	n	rc; Td-re	Se, Sf, Fec, Fer, *E* ●	+	[130]
		c	rc; Td-re	Sf, Fec, Fer	+	[161]
	*Neoseiulus cucumeris*	c	rc; Td, Tre (0) + Tp	Sel-a, Sf, Fec	+	[162]
	*Phytoseiulus longipes*	n	rc; Td-re	Sf, Fec, Fer, *E*, *IOBC*	+	[145]
			rc; Tre (4–31)		+ Tre (4)	
	*Phytoseiulus macropilis*	c	rc; Td-re	Sf, Fec, Fer	+	[161]
	*Phytoseiulus persimilis*	c	rc; Td, Tre (0) + Tp	Sel-a, Sf, Fec	+	[149]
		n	rc; Td-re	Se, Sf, Fec, Fer, *E* ●	+	[132]
oxymatrine	*Phytoseiulus longipes*	n	rc; Td-re	Sf, Fec, Fer, *E*, *IOBC*	−	[145]
			rc; Tre (4–31)		+ Tre (10)	
pyrethrum	*Amblyseius andersoni*	n	rc, Td-re	Se, Sf, Fec, *E*	−	[160]
	*Neoseiulus californicus*	n	rc; Td-re	Se, Sf, Fec, Fer, *E* ●	−	[130]
	*Phytoseiulus persimilis*	n	rc; Td-re	Se, Sf, Fec, Fer, *E* ●	−	[132]
rosemary oil	*Phytoseiulus persimilis*	c	rc, rc+/−; Td-re, Tre (0)	Sf, LC_50_ ●	+	[107]
soybean oil	*Phytoseiulus persimilis*	n	rc; Td-re	Se, Sf ●	+	[135]
+ fatty acids						
+ caraway oil						

c = commercial strains, n = native populations; rc = recommended concentrations; rc+/− concentrations higher/lower than rc; Td-re = direct (topical) treatment and residual exposure; Tre (0) = exposure to fresh residues; Tre (x–y) = exposure to residues aged from x to y days; Td = direct (topical) treatment and transfer to untreated surface; Tp = consumption of treated prey; Se = egg hatching; Sel-a = egg hatching and larval survival to the adulthood; Sl = larval survival; Sl−a = larval survival to the adulthood; Sf = adult female survival; Fec = fecundity (eggs produced by treated females); Fer = fertility (hatched eggs produced by treated females); *E* = the coefficient of toxicity [129]; *IOBC* = classification according the IOBC-WPRS guidelines [128]; ● comparative toxicity predator—prey; * Search in the literature (article written in English, published after the year 2000) was carried out by using combinations of several keywords in GoogleScholar, such as biopesticide, botanical, Tetranychidae, Phytoseiidae, compatibility, effects.

Investigation of the comparative toxicity between predators and their prey has an important place in this evaluation of the selectivity of bioacaricides. This type of investigation is the basis for assessing physiological selectivity [123]. The lower comparative toxicity to phytoseiids, however, does not itself ensure compatibility. If the bioacaricide significantly reduces the prey population, the lack of food may cause the decline and disappearance of the predator population, followed by an outbreak of the prey population [163]. Also, the consequences of approximately equal harmfulness of the recommended concentrations for predator and prey should be interpreted depending on the level of the harmful effects of a bioacaricide. Bergeron and Schmidt-Jeffris [138], for example, calculated abamectin selectivity indices for *N. fallacis*, *N. californicus* and *P. persimilis* to be zero. These values resulted from approximately equal but high harmfulness (96–100%) for both predators and their prey (*T. urticae*), so this bioacaricide is not rated as compatible. The predator–prey comparison using LC_50_ as the endpoint requires additional interpretation of the results, because significantly lower toxicity for the predator compared to the prey has practical significance if the LC_50_ for the predator is higher than the recommended concentration [152].

When investigating physiological selectivity, the phenomenon of acaricide resistance should be taken into account as well [123]. Regarding bioacaricides, the only example is resistance to abamectin and milbemectin, which is widespread in *T. urticae* populations [164] and also observed in other tetranychids [165,166].

Evaluations using demographic bioassays (Table 7) showed that the recommended concentrations of several tested bioacaricides reduce the population growth of phytoseiid mites. Lower concentrations can also cause reduction, as shown by the treatment of *Phytoseius plumifer* (Canestrini and Fanzago) with abamectin applied at LC_20_ against adult females [167]. On the other hand, the finding that low concentrations do not cause reduction [168,169] should be interpreted bearing in mind the recommended concentrations. Comparative demographic studies would allow for deeper insight, but there have been no examples of such research with bioacaricides to date.

**Table 7 insects-16-00095-t007:** Examples of evaluation of the compatibility of bioacaricides and phytoseiid mites in demographic bioassays (*r_m_* = the intrinsic rate of increase: ↓ = reduction, *ns* = non significant effect).

Bioacaricides	Phytoseiid Mites		Methodology	Effect on *r_m_*	References *
*Beauveria bassiana* GHA	*Phytoseiulus persimilis*	c	rc, ATlt, F_0_	↓	[170]
*Hirsutella thompsonii*	*Phytoseiulus longipes*	n	rc, ATlt, F_0_	↓	[145]
abamectin	*Phytoseiulus longipes*	n	rc, ATlt, F_0_	↓	[145]
	*Neoseiulus baraki*	n	rc, Flt, F_0_ F_1_	↓ F_0_	[171]
	*Phytoseius plumifer*	n	LC_10, 20_, Flt, F_1_	↓ LC_20_	[167]
milbemectin	*Amblyseius swirskii*	n	LC_5, 15, 25_, ATlt, F_1_	*ns*	[169]
azadirachtin	*Neoseiulus baraki*	n	rc, Flt, F_0_ F_1_	*ns*	[171]
	*Phytoseiulus longipes*	n	rc, ATlt, F_0_	↓	[145]
geraniol + citronellol	*Neoseiulus californicus*	c	LC_10, 20_, ATlt, F_1_	*ns*	[168]
+ nerolidol + farnesol					
oxymatrine	*Phytoseiulus longipes*	n	rc, ATlt, F_0_	↓	[145]

c = commercial strains, n = native populations; rc = recommended concentrations; ATlt = age-stage two-sex life tables; Flt = fertility life tables; F_0_ = treatment and assessment in F_0_ generation; F_1_ = treatment in F_0_ generation, assessment in F_1_ generation; F_0_ F_1_ = treatment in F_0_ generation, assessment in F_0_ and F_1_ generations; * Search in the literature (article written in English, published after the year 2000) was carried out by using combinations of several keywords in GoogleScholar, such as biopesticide, Tetranychidae, Phytoseiidae, life tables, compatibility, effects.

An alternative and less labor- and time-consuming approach for assessing effects at the population level is based on the calculation of the instantaneous rate of increase (*r_i_*) from the number of live individuals at the beginning and the end of a bioassay [172]. Using this approach, Tsolakis and Ragusa [135] compared the effects of a commercial mixture of vegetable and essential oils and potassium salts of fatty acids on the population growth of *T. urticae* and *P. persimilis*; the biopesticide significantly reduced the *r_i_* values of the former but not of the latter species. Lima et al. [173] observed significant reduction of the *r_i_* values following treatment of *Neoseiulus baraki* (Athias-Henriot) with abamectin, while Silva et al. [149] found no reduction in *Neoseiulus barkeri* Hughes treated with abamectin and azadirachtin.

The aforementioned examples show that life history traits have been the focus of evaluations of the sublethal effects of bioacaricides on phytoseiids. Few studies have focused on the evaluation of behavioral effects. Lima et al. [174] investigated repellence (avoiding acaricide without making direct contact with its residues) and irritancy (moving away from the treated area after making contact) of azadirachtin for *N. baraki* and found significant effects. The authors emphasized that these effects on walking behavior potentially reduce the exposure of predators, but also can cause their dispersal and decrease the efficiency of biological control. On the other hand, Bostanian et al. [136] and Beers and Schmidt-Jeffris [175] observed low levels of repellency of spinosad on *G. occidentalis*. Bioacaricides can also affect the feeding and reproduction behavior of phytoseiid mites. Investigations into the effects on *N. baraki* revealed that azadirachtin impaired copulation [176] and abamectin lowered the attack rate [177], while both bioacaricides impaired prey location [178]. These effects hinder predator–prey interaction and can compromise biological control.

The largest amount of data on the effects of acaricides on phytoseiids comes from laboratory bioassays. Therefore, the extrapolation of the results from the laboratory to the field is fundamentally important, especially for the IOBC-WPRS program for sequential testing. Translating laboratory results to field conditions has been a major challenge considering that laboratory bioassays most often do not take into account the complexity and variability of environmental factors, the heterogeneity of spray coverage in time and space or indirect effects through food supply and population dynamics [123,179]. In addition to these general limitations, IOBC-WPRS testing has been discussed critically regarding the realism of the “worst case” scenario and the rigid implementation of the trigger values for toxicity classification, which may affect the accuracy of predictions drawn from the laboratory bioassay alone [180,181,182]. Methodological issues aside, the usefulness of the IOBC database is limited by the need to evaluate the compatibility of pesticides (acaricides) with predatory species and strains important in local environments [123,179,182].

Investigating compatibility in greenhouse and/or field trials is necessary, and not only because of the problems that arise when extrapolating results from the laboratory. The compatibility of acaricides with phytoseiids should be tested and proven under realistic conditions of the complex interaction of various factors. Large-scale field trials enable an assessment of the long-term impact of operational factors (timing, procedures and amounts of acaricide application, predator augmentation, habitat management measures) and biological-ecological factors (trophic relationships and interactions, migrations, host plant features, refugia) on the dynamics of predator and prey populations and the effectiveness of biological control under variable environmental conditions. On the other hand, the complexity and variability of factors in the field can make it difficult to interpret the results and draw clear conclusions [123,180,182].

There have not been many examples of studies evaluating the compatibility of bioacaricides and phytoseiids in greenhouse and/or field trials. In some of these studies, the dynamics of both predator and prey populations were monitored in order to explore the possibility of combining predator activity and bioacaricide application. Greenhouse trials showed that mycopesticides could be successfully applied against *T. urticae* in combination with *P. persimilis* release on tomato plants [75] and *N. californicus* release on rose plants [183], as well as against mixed infestations of chrysanthemums with *T. urticae* and the western flower thrips, *Frankliniella occidentalis* (Pergande), in combination with *A. swirskii* and *N. cucumeris* [184]. In one of those trials, spinosad application at the rates recommended for thrips control was shown to be compatible with *P. persimilis* release against *T. urticae* on ivy geranium [185]. However, in another trial, the application of spinosad in an apple orchard was detrimental to *Kampimodromus aberrans* (Oudemans), a predator of *P. ulmi* [157].

In other studies, only the predator population was monitored. Jacas Miret and Garcia-Mari [186] rated abamectin as moderately harmful and azadirachtin as harmless to *Euseius stipulatus* (Athias-Henriot) following treatments in a citrus orchard. Castagnoli et al. [160] conducted field trials in an apple orchard in which azadirachtin and pyrethrum were applied twice on a bi-weekly basis. Azadirachtin did not affect the population of *A. andersoni*; pyrethrum significantly reduced the population density after the second treatment, but the population recovered in a few days. Miles and Dutton [187] investigated the effects of single and repeated applications of spinosad on phytoseiid populations in vineyards and apple orchards. Maximum reductions of 43% and 21% were found for *T. pyri* and *K. aberrans* populations in vineyards, respectively, and 41% was found for the *A. andersoni* population in apple orchards, after a single treatment of spinosad. Repeated applications caused a maximum reduction of 75% in *A. andersoni* population in apple orchards, and this effect was rated as harmful. On the other hand, de Andrade et al. [115] found no significant population reduction in *Iphiseoides zuluagai* Denmark and Muma (a predator of the citrus leprosis mite, *Brevipalpus yothersi* Baker) treated with oxymatrine in a commercial citrus grove.

Considering the advantages and limitations of different types of bioassays, a complementary approach to the evaluation of selectivity [123,160,180] that integrates laboratory and field data is needed as a sustainable solution for exploiting physiological and/or ensuring ecological selectivity. This integrative approach requires further research in the IPM context on an expanded range of evaluated bioacaricides and phytoseiid species and strains, both commercialized and native.

## 5. Bioacaricides—Potential and Perspectives

According to a long-term estimate [11], biopesticides will equalize with synthetic pesticides in terms of market size by the early 2050s. Whether the biopesticide sector will meet such great expectations depends on its response to various growth challenges. The most important challenge to the increase of the use of biopesticides in crop protection is the need for further improvements in their development, mass production, formulation and application [8,30].

Increasing the share of biopesticides on the global market can be achieved by the development of novel products, as well as by the improvement of products that are already on the market. As far as bioacaricides are concerned, new commercial products of microbial origin might be found among new strains and isolates of *Beauveria*, *Metarhizium* and other entomopathogenic fungi [188,189,190,191]. Candidates for new bioacaricides could also be based on bacterial species that are symbiotically associated with entomopathogenic nematodes (EPNs). Recent evaluations of cell suspensions (containing viable bacterial cells and toxins) and cell-free supernatants (containing only toxins) showed high acaropathogenic activity of several EPN-symbiotic species of the genera *Photorhabdus* and *Xenorhabdus* against spider mites [192,193,194]. The results of ongoing research have indicated new botanical sources of terpenoids/essential oils as potential bioacaricides [195,196,197]. Promising new ingredients for botanical bioacaricide products have also been found, such as coumarin scopoletin, which can be isolated from many plants [198], acetogenins from *Rollinia mucosa* (Jacq.) Baill (Annonaceae) [199], humilinolids from *Swietenia humilis* Zucc. (Lamiaceae) [200] and carlina oxide from *Carlina acaulis* L. (Asteraceae) [201].

The mass production of biopesticides depends on the availability and quality of biological resources. This issue is particularly important for botanicals, given that plant biomass has been a limitation to the commercialization of many potentially useful biopesticide agents. The shortage of biomass can be mitigated by the cultivation of plants containing bioactive compounds, as well as by the utilization of agro-industrial byproducts and waste. A good example of the latter is orange oil, obtained by cold pressing discarded citrus skins [40,94]. The inconsistent field efficacy of mycopesticides, which is a consequence of the sensitivity of entomopathogenic fungi to abiotic stresses and their naturally low virulence, can be overcome by genetic engineering [202,203].

The development of optimal formulations is required in order to ensure the stability and increase the efficacy of biopesticides. A recent approach to solving this issue has been through the micro- and nanoencapsulation of biopesticides using various matrices and carrier systems [204,205,206]. For example, Ebadollahi et al. [207] showed that loading of *Thymus eriocalyx* (Ronniger) Jalas and *Thymus kotschyanus* Boiss. and Hohen essential oils in a mesoporous material MCM-41 increased the persistence and toxicity of these oils to *T. urticae*, while Ahmadi et al. [208] found that the encapsulation of *Satureja hortensis* L. essential oil using chitosan nanoparticles improved its toxicity to this pest. Microencapsulated formulations increased the effectiveness of *B. bassiana* against *T. urticae*, and even low conidial concentrations of the fungus resulted in high mortality rates [209].

In addition to these technological challenges, the regulation of biopesticides and end-users’ perceptions about biopesticide effectiveness have been often emphasized as important impediments to their commercialization [8,40]. In order to overcome shortcomings in biopesticide regulation it is necessary to accelerate and simplify the processes involved in product registration. There is also a need for the global harmonization of biopesticide regulation [40,210,211]. More education, training and demonstrations are required to raise awareness among end-users about the effectiveness and benefits of biopesticides [8].

Further technological advances, legislation expediting registration and increased adoption, as well as the wider implementation of IPM and the expansion of organic agriculture, should contribute to the future growth of the biopesticide sector. However, the key driver of biopesticide success in the global market is and will remain profit [28,62]. An agrochemical company—large ones have recently moved into the biopesticide sector [8]—will produce biopesticides only if there is a profit to be gained. End-users (growers and farmers) will choose biopesticides only if they feel confident that the choice does not compromise their earnings.

## Figures and Tables

**Figure 1 insects-16-00095-f001:**
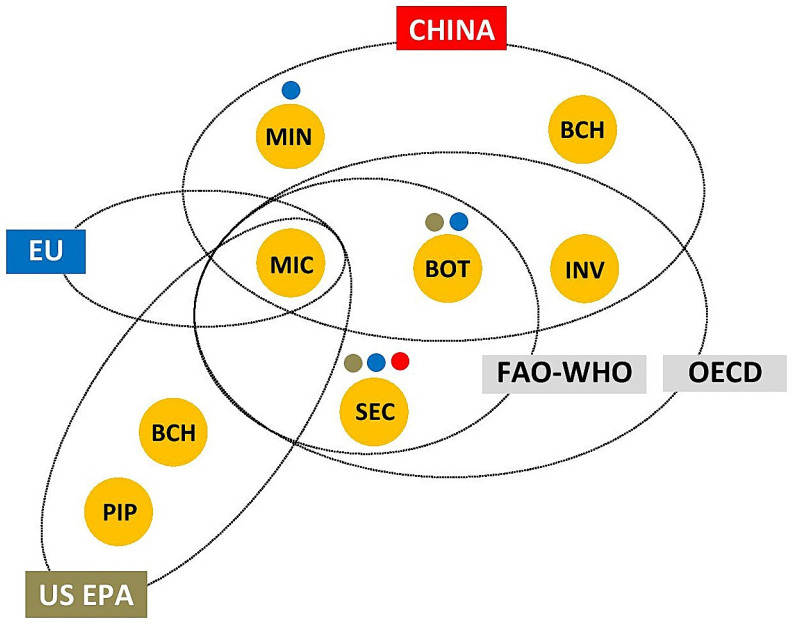
Biopesticides—definitions and classifications: INV = Invertebrates; BOT = Botanicals; SEC = Semiochemicals; MIC = Microbials (microbial pesticides); MIN = Microbial metabolites produced at the industrial scale (fermentation products of soil actinomycetes); PIP = Plant incorporated protectans; BCH = biochemical pesticides; OECD = Organisation for Economic Co-operation and Development, FAO = Food and Agriculture Organization; WHO = World Health Organization; EU = European Union; US EPA = United States Environmental Protection Agency; 

 EU legislation on plant protection products recognizes active substances that match BOT, SEC and MIN; 

 US EPA recognizes biochemical pesticides as a biopesticide class that matches SEC and BOT with non-toxic action; 

 China recognizes biochemical pesticides similarly to the United States’ EPA (plant-based growth regulators are included in BOT).

**Table 1 insects-16-00095-t001:** Examples of commercially available microbial acaricides—mycopesticides.

Fungal Species And Strains	Products	Manufacturers	Target Mites
*Beauveria bassiana*			
ATCC 74040	Naturalis L	Danstar Ferment (Hailey, ID, USA)	Tet, Tar, Ten
GHA	Mycotrol, Botanigard	Certis (Columbia, MD, USA)	Tet, Tar, Eri
PPRI 5339	Velifer	BASF Corporation (Florham Park, NJ, USA)	Pfm
	Broadband	BASF (Midrand, South Africa)	Tet
R444	Bb Protec, Eco-Bb	Andermatt (Howick, South Africa)	Tet
ESALQ-PL63	Boveril	Koppert Biological Systems (Piracicaba, Brazil)	Tur
IBCB-66	Bovebio	Biofungi (Eunapolis, Brazil)	Tur
	Beauveria Oligos	Oligos Biotecnologia (Piracicaba, Brazil)	Tur
	Eco-bass	Toyobo (Sao Paulo, Brazil)	Tur
Not specified	Boverin	Cherkasy Biozakhyst (Geronimovka Ukraine)	Tet
	Myco-Jaal	PCI (Mumbai, India)	Pfm
*Metarhizium brunneum* *			
F52/MA 43	Lalguard M52	Lallemand Plant Care (Milwaukee, WI, USA)	Pfm
*Isaria fumosorosea* **			
Apopka 97	PFR 97	Certis (Columbia, MD, USA)	Tet, Eri, Tar
FE9901	Isarid	Koppert Biological Systems (Howell, MI, USA)	Tet, Eri
PFA 011	Paecilomite	AgriLife (Hyderabad, India)	Pfm
*Lecanicillium lecanii*			
MCC 0058	Mealikil	AgriLife (Hyderabad, India)	Pfm
V-80	Biovert	Sibbiofarm (Berdsk, Russia)	Tet
*Lecanicillium muscarium*			
Not specified	Verticillin	Cherkasy Biozakhyst (Geronimovka, Ukraine)	Tet
*Hirsutella thompsonii*			
HT 019	Lancer	AgriLife (Hyderabad, India)	Tet, Eri
Not specified	Almite	IPL Biologicals (Gurugram, India)	Tet, Eri

This list was made by compiling data from several recent reviews of mycopesticides [46,63,64,68,70,71,72,73], official pesticide databases [25,69,74] and a number of manufacturer websites; Tur = *T. urticae*; Tet = Tetranychidae; Tar = Tarsonemidae; Eri = Eriophyidae; Ten = Tenuipalpidae; Pfm = (plant-feeding) mites; * formerly *M. anisopliae* var. *anisopliae* ** formerly *Paecilomyces fumosoroseus*.

**Table 2 insects-16-00095-t002:** Examples of commercially available botanical biochemical acaricides.

Active Ingredients	Products	Manufacturers	Target Mites
pyrethrum (pyrethrin I and II) ^1^	Pyrethrum	PelGar International (Alton, UK)	Tet
	Pyganic	MGK (Minneapolis, MN, USA)	Pfm
	Aphkiller	Beijing Kingbo Biotech (Beijing, China)	Tet
Azadirachtin ^2^	NeemAzal T/S	Trifolio-M (Lahnau, Germany)	Tet
	AzaGuard	BioSafe systems (East Hartford, CT, USA)	Tet, Eri
	Azatin	Certis (Columbia, MD, USA)	Tet, Eri
	Ozomite	Ozone Biotech (Faridabad, India)	Tet
	Neem Kil	Kilpest (Bhopal, India)	Pfm
	Ecotin	PJ Margo (Bangalore, India)	Tet
	Azamax	UPL (Ituverava, Brazil)	Tur
rosemary oil ^3^ (1.8-cineole) *	EcoTrol	KeyPlex (Winter Park, FL, USA)	Tet, Eri, Tar
+ geraniol			
+ peppermint oil ^4^ (menthol)			
orange oil ^5^ (*d*-limonene)	Prev-Am	Oro Agri (Cape Town, South Africa)	Pfm
tea tree oil ^6^ (terpinen-4-ol)	Eco-oil	OCP (Clayton, Australia)	Tur
+ eucalyptus oil ^7^ (1.8-cineole)			
cinnamon oil ^8^ (cynnamaldehyde)	Akabrown	GreenCorp Biorganiks (Coahuila, Mexico)	Tet
+ peppermint oil ^4^ (menthol)			
+ clove oil ^9^ (eugenol)			
+ oregano oil ^10^ (carvacrol)			
geraniol + citronellol	Mitexstream	Touchstone Env. Sol. (Sheridan, WY, USA)	Tet, Eri, Tar
geraniol + citronellol	Biomite	Arysta LifeScience (Cary, NC, USA)	Tet
+ nerolidol+ farnesol			
*α*-terpinene + *d*-limonene + *p*-cymene	Requiem Prime	Bayer Crop Science (Monheim, Germany)	Tet, Eri, Tar, Ten
matrine, oxymatrine ^11^	Matrine	Beijing Kingbo Biotech (Beijing, China)	Pfm
Veratrine ^12^	Marvee	Chengdu Newsun (Chengdu, China)	Tet
capsicum oleoresin ^13^ (capsaicin)	Captiva	Gowan (Yuma, AZ, USA)	Tet, Eri, Tar, Ten
+ canola oil ^14^ (oleic acid)			
+ garlic oil ^15^ (allyl sulfides)			
cottonseed oil ^16^ (oleic acid)	GC-Mite	JH Biotech (Ventura, CA, USA)	Pfm
+ clove oil ^9^ (eugenol)			
+ garlic oil ^15^ (allyl sulfides)			
castor oil ^17^ (ricinoleic acid)	TetraCURB Max	Kemin (Des Moines, IA, USA)	Tet
+ rosemary oil ^3^ (1.8-cineole)			
+ clove oil ^9^ (eugenol)			
+ peppermint oil ^4^ (menthol)			
canola oil ^14^ (oleic acid)	Neu1160	Neudorff (Emmerthal, Germany)	Tet
rapessed oil ^14^ (oleic acid)	Neu1160 I	Neudorff (Emmerthal, Germany)	Tet
safflower oil ^18^ (linoleic acid)	Suffoil	OAT Agrio (Tokyo, Japan)	Tet
+ cottonseed oil ^16^ (oleic acid)			

This list was made by compiling data from several recent reviews [10,23,26,40,94,95,96,97], official pesticide databases [25,69,74] and a number of manufacturer websites; Botanical sources: ^1^
*Tanacetum cinerariifolium* (Asteraceae); ^2^
*Azadirachta indica* (Meliaceae); ^3^
*Rosmarinus officinalis* (Lamiaceae); ^4^
*Mentha piperita* (Lamiaceae); ^5^
*Citrus sinensis* (Rutaceae); ^6^
*Melaleuca alternifolia* (Myrtaceae); ^7^
*Eucalyptus globulus* (Myrtaceae); ^8^
*Cinnamomum verum* (Lauraceae); ^9^
*Syzygium aromaticum* (Myrtaceae); ^10^
*Origanum vulgare* (Lamiaceae); ^11^
*Sophora flavescens* (Fabaceae), ^12^
*Veratrum nigrum* (Melanthiaceae); ^13^
*Capsicum annuum* (Solanaceae); ^14^
*Brassica napus*, *B. rapa* (Brassicaceae); ^15^
*Allium sativum* (Amaryllidaceae); ^16^
*Gossypium hirsutum* (Malvaceae); ^17^
*Ricinus communis* (Euphorbiaceae); ^18^
*Carthamus tinctorium* (Asteraceae); Tur = *T. urticae*; Tet = Tetranychidae; Tar = Tarsonemidae; Eri = Eriophyidae; Ten = Tenuipalpidae; Pfm = plant-feeding mites; * in brackets: the main active constituent.

**Table 3 insects-16-00095-t003:** (**a**) Evaluation of the compatibility of bioacaricides and phytoseiid mites according to the IOBC-WPRS standardized methods: the number of results listed in the database [125]; L = laboratory bioassays (○ initial toxicity bioassay; ● extended laboratory bioassay; ■ persistence bioassay); F = semi-field and field trials; T = total number of results (highlighted bold); (**b**) Evaluation of the compatibility of bioacaricides and phytoseiid mites according to the IOBC-WPRS standardized methods: IOBC-WPRS toxicity classification * [125]; L = laboratory bioassays, F = semi-field and field trials.

**(a)**																							
**Bioacaricides**	** *Aan* **		** *Ala* **		** *Asw* **		** *Ide* **		** *Kab* **		** *Nca* **		** *Ncu* **		** *Ppe* **		** *Ppl* **		** *Tpy* **		Σ		
	L	F	L	F	L	F	L	F	L	F	L	F	L	F	L	F	L	F	L	F	L	F	**T**
** Microbial **																							
*B. bassiana*											1●			1					8○●		9	1	**10**
*L. lecanii*															1●	1			1○	1	2	2	**4**
*I. fumosorosea*								1								1						2	**2**
*M. brunneum*						1																1	**1**
** Biochemical **																							
abamectin			1○			1					2■				1●		1●		7●	2	12	3	**15**
milbemectin																			7○●		7		**7**
spinosad									1●		1■								10○●		12		**12**
azadirachtin															2●				2○●	1	4	1	**5**
fatty acids	1○														1○				2○●		4		**4**
rapeseed oil															1●				3●		4		**4**
thymol																			1○		1		**1**
	1		1			2		1	1		3	1		1	6	2	1		41	4	55	10	**65**
**(b)**																							
**Bioacaricides**	** *Aan* **		** *Ala* **		** *Asw* **		** *Ide* **		** *Kab* **		** *Nca* **		** *Ncu* **		** *Ppe* **		** *Ppl* **		** *Tpy* **				
	L	F	L	F	L	F	L	F	L	F	L	F	L	F	L	F	L	F	L	F			
** Microbial **																							
*B. bassiana*											2			1					2-1				
*L. lecanii*															1	1			1	2			
*I. fumosorosea*								1								1							
*M. brunneum*						1																	
** Biochemical **																							
abamectin			3			2					3-1 *a*				4		4		4-2	1			
milbemectin																			4-1				
spinosad									4		1 *b*								4-1				
azadirachtin															4-1				3-1	1			
fatty acids	3														1				3-1				
rapeseed oil															3-1				3-1				
thymol																			1				

*Aan* = *Amblyseius andersoni*; *Ala* = *A. largoensis*; *Asw* = *A. swirskii*; *Ide* = *Iphiseius degenerans*; *Kab* = *Kampimodromus aberrans*; *Nca* = *Neoseiulus californicus*; *Ncu* = *N. cucumeris*; *Ppe* = *Phytoseiulus persimilis*; *Ppl* = *Phytoseius plumifer*; *Tpy* = *Typhlodromus pyri*. *Aan* = *Amblyseius andersoni*; *Ala* = *A. largoensis*; *Asw* = *A. swirskii*; *Ide* = *Iphiseius degenerans*; *Kab* = *Kampimodromus aberrans*; *Nca* = *Neoseiulus californicus*; *Ncu* = *N. cucumeris*; *Ppe* = *Phytoseiulus persimilis*; *Ppl* = *Phytoseius plumifer*; *Tpy* = *Typhlodromus pyri*. * Laboratory bioassays: 1 = harmless (*E* < 30%), 2 = slightly harmful (*E* = 30–80%), 3 = moderately harmful (*E* = 80–99%), and 4 = harmful (*E* > 99%); Field trials: 1 = harmless (*E*< 25%), 2 = slightly harmful (*E* = 25–50%), 3 = moderately harmful (*E* = 50–75%), and 4 = harmful (E > 75%); *E* = the total effect (%); Underlined toxicity classes refer to the persistence bioassay: *a* = residue aged 5 days (3) to 15–30 days (1), *b* = residue aged 5 days.

**Table 4 insects-16-00095-t004:** Examples of evaluation of the compatibility of microbial bioacaricides (mycopesticides) and phytoseiid mites in various laboratory bioassays; Comp = compatibility conclusion: + positive (compatible); − negative (not compatible).

Mycopesticides	Phytoseiid Mites		Methodology		Comp	References *
			Exposure	Endpoints		
*Beauveria bassiana* GHA	*Amblyseius swirskii*	c	rc; Td-re	Sf, Fec, Fer	+	[144]
*Beauveria bassiana* ATCC 74040	*Neoseiulus californicus*	n	rc; Td-re	Se, Sf, Fec, Fer, *E* ●	−	[130]
	*Phytoseiulus persimilis*	n	rc; Td-re	Se, Sf, Fec, Fer, *E* ●	+	[132]
*Hirsutella thompsonii*	*Phytoseiulus longipes*	n	rc; Td-re	Sf, Fec, Fer, *E*, *IOBC*	+	[145]
			rc; Tre (4-31)		+ Tre (10)	

c = commercial strains, n = native populations; rc = recommended concentrations; Td-re = direct (topical) treatment and residual exposure; Tre (x-y) = exposure to residues aged from x to y days; Se = egg hatching; Sf = adult female survival; Fec = fecundity (eggs produced by treated females); Fer = fertility (hatched eggs produced by treated females); *E* = the coefficient of toxicity [129]; *IOBC* = classification of toxicity according the IOBC-WPRS guidelines [128]; ● comparative toxicity predator—prey; * Search of the literature (article written in English, published after the year 2000) was carried out by using combinations of several keywords in GoogleScholar, such as biopesticide, mycopesticide, Tetranychidae, Phytoseiidae, compatibility, effects, etc.

## Data Availability

No new data were created or analyzed in this study. Data sharing is not applicable to this article.

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
