# Peer review of "Bioacaricides in Crop Protection—What Is the State of Play?"

_insects, 2025, doi:10.3390/insects16010095_

Round 1
Reviewer 1 Report
Comments and Suggestions for Authors
I have reviewed the manuscript entitled "Bioacaricides in crop protection – what's the state of play?" The authors provide an overview of the properties, effects, and uses of contemporary bioacaricides used to control plant-feeding mites and their compatibility with Phytoseiidae family predatory mites.
It is a very interesting document with valuable information for many entomologists and acarologists involved in pest control using biological products.
It is recommended that the authors incorporate the methodology of collecting all the information, indicating the databases they visited and the techniques used to collect the information.
Other suggestions.
Line 229. Separate the words "fromthe" should be "from the"
Line 266. Separate the words "Certainproducts" should be "Certain products"
Lines 538-539. Between those lines, parentheses are missing or extra.
In Table 4. Are Beauveria bassiana GHA, Amblyseius swirskii, and c in bold?
Line 668. Change "(" to "["
Line 677. Delete "rate" because it is repeated.
Within the text, in some references they use "&" and in others "and", please homogenize. For example, in lines 652, "Bergeron & Schmidt-Jeffris [138]" and line 742, "Miles and Dutton [187]".
Author Response
It is recommended that the authors incorporate the methodology of collecting all the information, indicating the databases they visited and the techniques used to collect the information.
The databases and references are indicated in the tables. Literature search (article written in English, published after the year 2000) was carried out by using combinations of several keywords in GoogleScholar, such as biopesticide, botanical, microbial, Tetranychidae, Phytoseiidae, effects, etc.
Other suggestions.
Line 229. Separate the words "fromthe" should be "from the"
Line 266. Separate the words "Certainproducts" should be "Certain products"
Lines 538-539. Between those lines, parentheses are missing or extra.
In Table 4. Are Beauveria bassiana GHA, Amblyseius swirskii, and c in bold?
Line 668. Change "(" to "["
Line 677. Delete "rate" because it is repeated.
Within the text, in some references they use "&" and in others "and", please homogenize. For example, in lines 652, "Bergeron & Schmidt-Jeffris [138]" and line 742, "Miles and Dutton [187]".
Corrected.
Reviewer 2 Report
Comments and Suggestions for Authors
This is a nice, well pre´pared and very informative manuscript. I present some suggestions for improvement in the attached file, almost all of which refer to formating. Only two "conceptual" points deserve to be highlighted:
a) the used concept of "natural enemy"
b) the ending part, which IN MY VIEW seems a light bit too sober. Without wanting to suggest exagerations, perhaps it could be a little more enthusiastic.

Author Response
I present some suggestions for improvement ...almost all of which refer to formating.
Accepted.
Only two "conceptual" points deserve to be highlighted:
a) the used concept of "natural enemy"
It has been clarified by replacing the term natural enemies by the term parasitoid and predator
b) the ending part, which IN MY VIEW seems a light bit too sober. Without wanting to suggest exagerations, perhaps it could be a little more enthusiastic.
In my view, the ending part is realistic.